# Applications and outcomes of implementing telemedicine for hypertension management in COVID-19 pandemic: A systematic review

**Mohammad Hosein Hayavi-haghighi[1], Abdullah Gharibzade[2‡], Niloofar Choobin[3‡], Haniyeh Ansarifard [3] ***

**1** Department of Health Information Technology, School of Allied Medical Sciences, Hormozgan University of Medical Sciences, Bandar Abbas, Iran, **2** Department of cardiology, School of medicine, Tobacco and Health Research Center, Hormozgan University of Medical Sciences, Bandar Abbas, Iran, **3** Faculty of Para-medicine, Hormozgan University of Medical Sciences, Bandar Abbas, Iran

☯ These authors contributed equally to this work.
‡ AG and NC also contributed equally to this work.
* haniyehansarifard99@gmail.com

**Data Availability Statement:** All relevant data are within the manuscript and its Supporting Information files.

## Abstract

### Introduction

COVID-19 presented a significant challenge for patients with hypertension in terms of access to care. However, telemedicine offered the healthcare system opportunities that had previously been underutilized. Therefore, this study aims to systematically review the applications and outcomes of telemedicine for hypertension management during the COVID-19 pandemic.

### Method

A structured search was conducted in accordance with PRISMA guidelines across multiple databases, including PubMed, Cochrane, Web of Science, and Scopus. The search was limited to studies published from December 2019 until May 2023, resulting in a total of 3727 studies. After quality appraisal using the CASP checklists version 2018, 29 articles were included in the final review. Data analysis was performed using thematic analysis.

### Results

Most of the studies reviewed had used the proprietary platforms (N = 14) and 11 studies had used public platforms such as social messengers or email. Also 9 studies relied on phone calls (N = 9) to record and transmit the clinical data. Some studies had applied two different approaches (proprietary/public platforms and phone). six articles (20.7%) focused only on hypertension control, while 23 articles (79.3%) examined hypertension as a comorbidity with other diseases. Also, the study identified 88 unique concepts, 15 initial themes, and six final themes for outcomes of using telemedicine for hypertension management during the COVID-19 pandemic. These themes include BP control, BP measurement and recording, medication management, mental health, care continuity and use and acceptance.

**Funding:** The author(s) received no specific funding for this work.

**Competing interests:** NO authors have competing interests.

## Conclusion

Telemedicine provides patients with hypertension with the opportunity to engage in medical consultations in a more convenient and comfortable manner, with the same validity as in-person visits. Telemedicine facilitates the creation of a connected network to support patients with high BP at any time and in any location. Limitations and issues may arise due to patients and healthcare staff's unfamiliarity with telemedicine. These issues can be resolved through the ongoing use and continuous feedback.

## 1. Introduction

The use of telemedicine technologies, such as mobile phones and other audio-visual smart devices, has significantly transformed disease management and improved access to ambulatory and subspecialty care for patients in disadvantaged populations [1–3]. Mobile health (M-health), as a subset of telemedicine, involves the use of mobile phones and other wireless technologies in health care. Wireless technologies in M-health enable easier access to caregivers, better disease monitoring, and ultimately higher health status [4]. Recent developments in M-health have introduced new opportunities for improving access to healthcare, enabling better self-management of chronic diseases, facilitating access to information, promoting healthy eating habits and increasing physical activity levels [4]. The COVID-19 pandemic has led to a rapid expansion in the use of telemedicine and M-health [5]. Patients often opt for remote appointments via communication technology when a physician is unavailable or during pandemics such as COVID-19 [1]. These appointments provide comparable outcomes to face-to-face care and result in higher levels of satisfaction for both healthcare professionals and patients [5]. During the COVID-19 pandemic, the use of smartphones and low-cost wireless devices linked to smartphone applications has facilitated the collection and dissemination of accurate data among patients and physicians, thereby limiting reporting errors [6].

The main purpose of telemedicine is to educate of consumers about preventive health care, disease monitoring, treatment support, epidemic tracking, and chronic disease management [7, 8]. Advanced technologies such as Bluetooth and motion detection sensors (such as accelerometer and gyroscope) have led to the creation of numerous applications, particularly for chronic diseases including diabetes [4], heart disease [9], kidney problems [10], and especially hypertension [4]. This allows patients with chronic diseases to access medical services more easily than in the pre-technology era [11].

Hypertension, also known as high blood pressure (BP), is a significant public health concern and the most common chronic disease in primary care [12]. It affects over one billion people worldwide [10]. In 2020, researchers reported a decline in both awareness of condition and effective management of hypertension among patients over the past decade [13]. The COVID-19 pandemic has worsened the problem, as the number of referrals of patients with hypertension to health care facilities have decreased by over 25% [14]. Failure to manage hypertension can result in various complications, including heart failure, cardiovascular disease, stroke, kidney disease, and ultimately death [10, 15].

However, hypertension can be effectively controlled with medication, diet and lifestyle changes, [16]. Patients with non-communicable diseases, such as hypertension, are particularly vulnerable to COVID-19. Therefore, disease management is of paramount importance during this pandemic [3]. additionally, the pandemic has posed challenges to the care of patients with

chronic diseases, particularly hypertension [17]. These patients were at risk of not receiving essential care due to quarantine limitations [10]. Therefore, the health care system should develop new approaches to prevent, diagnose and treat of non-communicable diseases [18]. One such strategy is the use of telemedicine platforms, which can help address important gaps in hypertension management, including access to care, medication adherence and patient engagement with an efficient, effective and patient-centered approach [14]. the use of telemedicine is not without its own set of challenges. The first of these is related to the quality of the internet connection and the subsequent difficulties encountered when implementing technology. The second is related to lack of physical examinations [19]. In addition, issues such as individual, racial and ethnic factors, as well as a lack of health and digital literacy must be considered [20]. Furthermore, the lack of full health insurance coverage and reimbursement, constitutes a significant barrier to the wider adoption of telemedicine [21].

Telemedicine and M-health have become a practical and attractive solutions for improving hypertension management and saving time and money for patients, caregivers and physicians. They are also technological solutions for preventing the spread of disease, screening patients, improving the quality of care, and providing real-time follow-up for patients with hypertension [6]. Telemedicine offers the opportunity to control and prevent the progression of the disease by continuously monitoring BP, encouraging medication adherence, promoting proper diet, and lifestyle changes [22]. Studies conducted during the COVID-19 pandemic have shown the effectiveness of telemedicine in controlling hypertension, indicating the understanding of society, physicians, and patients of its potential to overcome the challenges associated with the pandemic. These studies offer valuable insights for developing effective strategies to promote the regular and meaningful use of telemedicine. The Covid-19 pandemic presented a significant challenge for patients with hypertension in terms of accessing to care. But on the other hand, telemedicine has opened up opportunities for the healthcare system that were previously underutilized. Therefore, the aim of this study is to conduct a systematic review of the applications and outcomes of telemedicine for managing hypertension during the COVID-19 pandemic.

## 2. Material and methods

The study followed to the PRISMA (Preferred Reporting Items for Systematic Reviews and Meta-Analyses) guidelines. These guidelines provide a minimal set of evidence-based elements for reporting systematic reviews and meta-analyses and include a four-phase flow chart that has been approved as a standard by health science organizations and journals [23].

### 2.1 Search strategy

To conduct the search, we first extracted the primary keywords utilizing Medical Subject Headings (MeSH) and determining all combinations using the Boolean operators "OR" and "AND". These key words were hypertension OR high blood pressure OR blood pressure AND Telehealth OR telemedicine OR mobile health OR mhealth OR virtual health AND COVID-19 OR SARS-COV-2 OR 2019 Novel Coronavirus (S1 Appendix).

### 2.2 Information sources

Four research databases were examined to identify relevant sources: PubMed (Medline), Cochrane, Web of Science, and Scopus.

The search period began on December 2019, which coincides with the official announcement of the COVID-19 pandemic, and concluded on May 10, 2023, marking the beginning of this study.

## 2.3 Selection process

The Mesh protocol was utilized to conduct a comprehensive search across all relevant databases. Subsequently, the findings were entered into Endnote software version 20. To eliminate any duplicates, we used the software's duplicate elimination feature, followed by manual removal of any remaining duplicates by the first author. We then reviewed the article titles to discard any that did not align with the study's objective. The articles' abstracts underwent scrutiny to exclude any irrelevant ones. Two reviewers, H A and N CH, evaluated each study based on the title and abstract. They included approved studies and excluded rejected ones. If only one reviewer thought a study was good enough, the third reviewer (MH) decided. We then looked at all the articles that met the criteria and wrote up our findings in line with the PRISMA 2020 guidelines (Fig 1).

## 2.4 Eligibility criteria

The inclusion criteria for this study were as follows: studies targeting hypertension treatment in individuals aged 18 years and above by telemedicine modalities. Only RCTs, cohort, or observational studies were considered. It is of the utmost importance that these criteria are adhered to in order to guarantee an impartial and equitable analysis.

   The exclusion criteria were as follow: pregnancy-induced hypertension, telemedicine provision exclusively for treating patients with COVID-19, qualitative and survey studies, case reports, designs and models, and both narrative and systematic reviews. (Naturally, we employed the references of these studies during backward screening process.)

## 2.5 Quality assessment

In order to assess the quality of the included studies, we employed the Critical Appraisal Skills Program (CASP) checklists, which include 11 questions for assessing RCT studies and 12 questions for evaluating cohort and observational studies. In order to be included in the analysis, studies were required to achieve a minimum score of 60% for the full text.

## 2.6 Data extraction and synthesis

The articles that underwent full-text review were analyzed, and data elements were extracted using a pre-designed table.

   This table provides details on 29 reviewed studies. Information included in the table is study objectives, sample size, population, and training period and comorbidities. Additionally, thematic analysis was employed to determine the outcomes of using telemedicine for hypertension management. This approach was used to identify, extract, and summarize themes from the studies included in literature reviews. The process entailed four phases: coding data, extracting unique concepts, grouping them into initial themes, and finally developing final themes.

## 3. Findings

Following the initial search, 3727 articles were retrieved. After reviewing the titles and abstracts, 33 articles were examined in full text. Of these, four full texts were unavailable, leaving 29 articles included in the present study (S1 Checklist). Additionally, all qualifying studies achieved the necessary quality score (S2 Appendix).

   The majority of the studies (N = 15, 51.8%) were cohort studies that compared patients whose hypertension was managed using telehealth with a control group. A small number of clinical trials (N = 3, 10.3%) were conducted due to the epidemic and quarantine conditions,

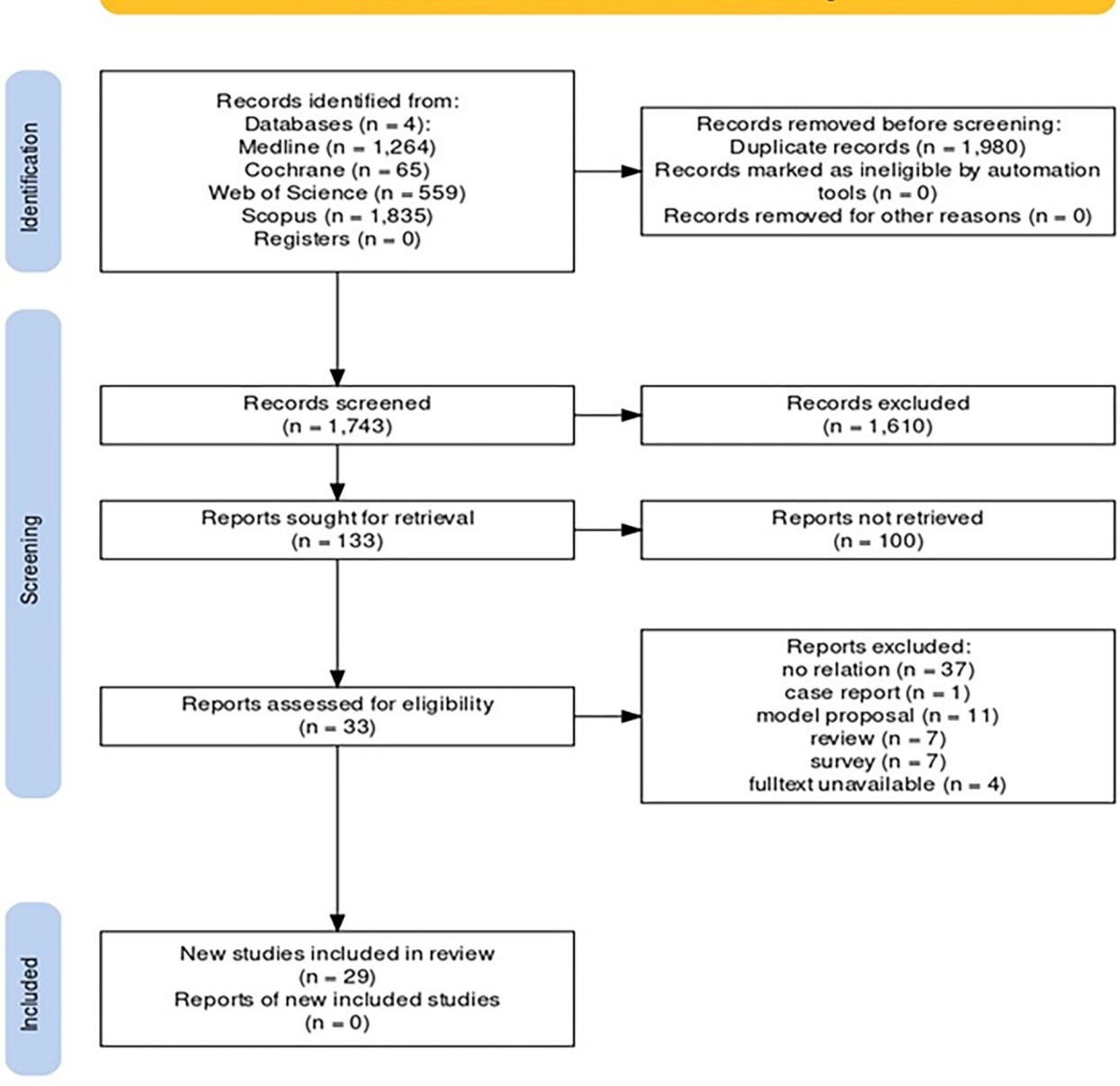

**Fig 1. PRISMA flow chart of the study selection process.**

as well as the desire of physicians and patients to avoid in-person visits to prevent the transmission of infection. About one-third of the reviewed studies were observational articles (N = 11, 37.9%). Of the studies analyzed, 14 (48.3%) were conducted in 2022. The sample size of the studies ranged from 20 to 1,223,560 visits, and the age of the patients ranged from 18 to 88 years. Cohort studies lasted from four days to four and a half years, observational studies lasted from six months to six years, and RCTs lasted from three to four months (Table 1).

Out of the 29 studies reviewed, 6 articles (20.7%) focused only on hypertension control, while 23 articles (79.3%) examined hypertension as a comorbidity with other diseases. Three distinct strategies were employed to facilitate communication between providers and patients.

**Table 1. Characteristics extracted from the included studies.**

| Name/Year/Country | Purpose | Design Population (age/sex) | N/Duration | App / Modality | Training content/ method | comorbidities |
|---|---|---|---|---|---|---|
| Alexander [24] 2020 USA | To quantify national changes in the volume, type, and content of primary care delivered during the COVID-19 pandemic, especially with regard to office-based vs telemedicine encounters. | Observational Age (<19_66) | 1223560 (Visit) 15 Months | NS | NS | Cholesterol /cardiovascular diseases |
| Alsaqer [10] 2022 Jordan | To examine the effects of a public health nursing intervention plus m-health applications for hypertension management on enhancing the self-care, systolic and diastolic of BP, and Quality of life in older adults during the lockdown period | RCT Age (55_88) F(43.6%) | 110 3 Months | My Heart / Pill Reminder/ Breathe Easy/ Steps | About high BP/ using a home BP monitor/ Life style point | NS |
| Armitage [25] 2022 Uk | To examine the process and Performance of ABPM when delivered remotely, using FMEA and a quantitative analysis that compared ambulatory blood Pressure data from participants receiving remote ABPM appointments, versus ambulatory blood pressure data from participants receiving face-to-face ABPM appointments. | Cohort Age (18_80) | 65 54 days | Nye Health / Ecg Partner Application Kardia/ Alivecor Kardiamobile Ecg Device | How to record ABPM An introductory session in FMEA was conducted for the panel, followed by six weekly sessions | Diabetes |
| Barake [26] 2022 Chile | To evaluate the trends In hypertension treatment and control rates in Chile during five years (2017–2021) | Observational Age (15–65<) | - 5 Years | Cvhp/ Heart | NS | NS |
| Blackstone [1] 2022 Charlottesville, USA | To use of telemedicine in refugee primary care | Cohort Age (18–65<) | 16386 11 Months | My Chart/ Cyracom | NS | diabetes |
| Bruns [3] 2024 USA | To evaluated the clinical impact of controlling hypertension by comparing in-person pharmacist visits to tele pharmacy visits in patients with hypertension. | Cohort Age (18) F (62.3%) | 54 Months | Telephone Call | How to use automated BP devices | diabetes mellitus type 2 /hyperlipidemia |
| Burgos [27] 2020 Argentina | To assess the feasibility and acceptability of virtual visits in patients with Heart Failure, pulmonary hypertension and heart transplant. | Cohort Age (15_65) F (11.7%) | 334 30 Days | Virtual Platforms | how to record vital signs and weight and detect signs of congestion, and written information about care | pulmonary hypertension/ heart transplant/ |
| Franco [28] 2022 Brazil | To Evaluate the Impact of a Telehealth intervention on metabolic outcomes and self-perceptions of the patients regarding their management of diabetes during the COVID-19 pandemic. | RCT Age (18<) F (50%) | (150)86 4 Months | Free Website | NS | diabetes mellitus |
| Gallardo-Rincón [18] 2022 Mexico | To investigate the disease profile of the population screened using MIDO covid in Mexico, and to identify diagnosis gaps in the continuum of care from screening to control | Observational Age (20–60<) F (39.5%) | 58,277 6 Months | Mido Covid | NS | type 2 diabetes |

*(Continued)*

**Table 1.** (Continued)

| Name/Year/ Country | Purpose | Design Population (age/sex) | N/ Duration | App / Modality | Training content/ method | comorbidities |
|---|---|---|---|---|---|---|
| Gibson [29] 2023 Ireland | To examine the outcomes (lifestyle, Risk factor, therapeutic goals and quality of life) of a digital program (Croí Mysláinte) at 12 weeks and at 6-month follow- up. | Observational Age (≥18) F (19%) | 55 6 Months | Digital Cr Program (Croí Mysláinte) | to guide self-measurement of blood pressure. | coronary heart disease |
| Girerd [30] 2022 Paris (France ( | To evidence of a blood pressure reduction during the COVID-19 pandemic and associated lockdown period: insights from e-health data | Cohort Age (<60) F(18.9) | 2273 5 Months | Withings | NS | NS |
| Hernández- Galdamez, [17] 2021 Guatemalan | To describe the approach used to monitor study participants during the COVID-19 pandemic and compare data obtained during phone calls for intervention and control group participants. | Cohort Age(63) F (71.68%) | 1384 4 Months | Telephone Call | correct method of blood pressure measurement | Obesity/ Dyslipidemia/ Diabetes Mellitus/ Depression |
| Iliuta [31] 2022 Romania | To examine the COVID-19 impact on the follow-up of patients with dilated cardiomyopathy and to establish the advantages of multiparametric home monitoring. | Observational Age(75<) F (39.4%) | 142 3 Years | Dedicated Application/ Short Message / Email/ Whatsapp | NS | COPD/ Chronic kidney disease/ DM |
| Kang [32] 2021 Changchun (China) | To confirm the effectiveness of remote management using the mobile phone WeChat app on comprehensive management of diabetes mellitus during the COVID-19 epidemic | Rct Age (<18_ >70) F (36.7%) | 160 3 Months | Wechat | about the necessity of exercise; monitoring guidelines and goals; and psychological knowledge | diabetes mellitus |
| Lee [33] 2023 Boston (USA) | To tested the hypothesis that an entirely remote hypertension management program could be adapted and strengthened to achieve successful BP control during the COVID-19 pandemic. | Observational Age (26–81) F (57%) | 1256 12 months | Umbrella Electronic Portal | BP measurement technique | Atherosclerotic cardiovascular/ Type 2 diabetes mellitus |
| Li, X. Y. [34] 2022 Shanghai, (China) | To evaluate the pharmaceutical telemedicine care service in patients with existing hypertension, as compared to usual care | Cohort Age (≥ 18) F (37.8%) | 352 6 Months | WeChat | how to: Measure BP at home; Enter and view data; and develop a BP management plan | Diabetes mellitus, CHD/ |
| Moreira [35] 2021 São Paulo (Brazil) | To assessing the short-term results of measures adopted in response to COVID-19 pandemic by using telemedicine in the following-up of patients at high cardiovascular risk. | Cohort Age (55–75) F (38%) | 240 4 Days | Telephone Call | NS | Diabetes/coronary artery disease/MI |
| Nacak [4] 2023 Turkiye | To determine the treatment compliance and metabolic control levels of T2DM patients who are remotely monitored with m-health technologies used for the 1st time in our country, and to evaluate the effects of being in this system on patients. | Observational Age (25–55) | 86 4 months | Metaclinic / Dm4all | NS | T2DM |
| Omboni [6] 2021 Italy | To present the experience based on a telehealth platform used at scale to manage chronic disease patients in the Italian community | Observational Age (40–76) F (55.46%) | 13,613 6 months | Telehealth Web Platform/ Mobile App/ Email | NS | NS |

(*Continued*)

**Table 1.** (Continued)

| Name/Year/ Country | Purpose | Design Population (age/sex) | N/ Duration | App / Modality | Training content/ method | comorbidities |
|---|---|---|---|---|---|---|
| Onea [36] 2023 Romania | To evaluate the changes in the metabolic control during the COVID-19 pandemic in T2DM subjects & whether care through telemedicine had significant impact on maintaining the glycemic control. | Observational F (48.2%) | 328 6 Years | Telephone/ Email | NS | T2DM |
| Park [5] 2022 Korea | To investigate the effects of video-based telehealth services using a mobile personal health record (PHR) app for vulnerable workers with metabolic risk factors. | Observational Age (≥ 19) F (59.0) | 117 4 months | Mobile Phr App | taught to use the mobile PHR app to connect with healthcare professionals | Fasting blood glucose/ HDL cholesterol/ Triglyceride |
| Ploux [37] 2021 France | To compare healthcare use, physiological variables, and HF decompensations during 1 month before and during the first month of the first French national lockdown for COVID-19 among patients undergoing remote monitoring | Cohort Age(>_80) F (16%) | 51 2 months | Remote Monitoring System (Careline Solutionstm, Me ´Rignac, France) | heart failure education program | /Diabetes mellitus / Dyslipidaemia |
| Russo [11] 2022 Italy | To describe the use of telemedicine during the COVID-19 pandemic in the year 2020, and to examine its impact on the volume activity and quality of diabetes care in the large cohort of t2dm patients. | Cohort Age (58_82) F (41.4%) | Face: 364,898 T:46,424 - | Telephone | NS | T2DM |
| Sreedhara [38] 2022 USA | To examine telehealth use among patients with hypertension, with a focus on populations who experience barriers to care & explore the effect of telehealth on blood pressure outcomes. | Observational F (43.7%) | Arcare = 574 Terros = 986 29 months | Telephone & Video Conference | NS | NS |
| Steiner [12] 2023 USA | To care for hypertension during the COVID-19 pandemic in an integrated health care system | Cohort Age (≥ 18) F (51.2%) | 64766 24 months | Telephone | NS | NS |
| Taylor [14] 2022 USA | To assess the effectiveness of telemedicine video visits in the management of hypertensive patients at home during the first year of the COVID-19 pandemic. | Cohort Age(40_60) F (48.3%) | 569 12 months | Digital Platform (Mobile Or Computers Devices | Providers received training in telemedicine manner and communication | Diabetes /CKD / Dyslipidemia / Ischemic heart disease/ COPD |
| Tierney [39] 2023 USA | To examine the association of care continuity with diabetes and hypertension care quality before and during COVID-19 and the mediating effect of telehealth. | Cohort Age(57.8) F (58.05%) | 20792 12 Months | NS | NS | T2DM CHD/ /Depression |
| Walker [40] 2021 UK | To evaluate outcomes from a six-session, 10-week program which, due to COVID-19 restrictions, was delivered on zoom rather than face to face as originally intended. | Cohort Age (≥18) F (85%) | 20 10 Weeks | Zoom/ Nhs43 | education on physical activity, intermittent fasting, gut health, stress management, sleep and behavior change | T2DM |
| Ye, S [41] Anstey 2022 USA | To examine the association between telemedicine visits and the failure to meet the controlling hypertension (BP) quality measure from the centers for Medicare & Medicaid services. | Cohort Age(18_85) F(52%) | 32,727 11 Months | Telephone | NS | cardiovascular disease / T2DM |

The majority of studies reviewed employed exclusively proprietary platforms (n = 11) or in conjunction with public platforms (n = 3). A second approach involved the use of public platforms, such as public social messengers or email, exclusively (n = 5) or in conjunction with proprietary platforms(n = 3) or phone calls (n = 5). The final strategy involved the use of telephone calls, either exclusively(n = 6) or as an adjunct to public platforms (n = 3). Two studies did not provide any information regarding the communication mechanism. It is notable that six studies employed two strategies concurrently (public platforms in conjunction with proprietary platforms or telephone calls).

15 studies (51.7%) utilized both face-to-face and virtual meetings, including video calls, to regularly compare in-person blood pressure readings with those taken remotely by the patients themselves. The study identified 88 unique concepts, 15 initial themes, and six final themes related to the outcomes of using telemedicine in hypertension management (Table 2). 21 articles reported the effect of telemedicine on hypertension control, seven articles discussed measuring and recording BP, nine focused on medication management, three addressed mental health, five discussed care continuity, and finally 15 examined the acceptance and use of telemedicine among hypertension patients.

## 4. Discussion

This study investigates the applications and outcomes of telemedicine in managing hypertension amidst the COVID-19 pandemic. As health systems around the world struggle to meet user needs, telemedicine has emerged as the most effective way to reduce risks for healthcare workers and patients during the pandemic. Although it was previously used to some extent in developed countries, its use has now expanded exponentially. The increasing popularity of this healthcare approach is due to its prioritization of safety and convenience [42].

Our results indicate that countries with high internet and smartphone penetration have conducted the most cases of telemedicine implementations for hypertension management. This suggest that health policy makers can utilize existing strong technical infrastructure to ensure the continuous delivery of services to their target population. In general, the use of telemedicine, and mobile health in particular, to deliver health services during the COVID-19 pandemic experienced explosive growth [43]. However, this growth was primarily observed in societies that had already stablished the necessary technical infrastructure, legal framework, and administrative rules. This potential has also been utilized for managing hypertension, enabling the provision of care, the continuation of care and follow-up care in a non-attendance setting [44].

There are three different approaches in implementing telemedicine for hypertension management. The first approach involves using a proprietary application or platform for continuous monitoring and care of hypertension. The second approach involves using commonly available communication applications, such as email, WhatsApp, WeChat, or suitable remote communication programs, to effectively communicate with participants and transfer data, training and/or providing necessary instructions. The third approach relied on telephone communication. It should be noted that this approach may not be appropriate in all situations and requires further evaluation.

To understand how each method affects the study's results, we analyzed the data. This analysis showed that proprietary platforms focus on final clinical outcomes, while studies using telephone calls focus on intermediate outcomes. Studies based on public platforms also fall somewhere in the middle.

Of the 21 studies on blood pressure control, 11 used proprietary systems, 6 studies used public systems, and 4 studies used telephone. This indicates that in 75.6% of studies using

**Table 2. Outcomes of telemedicine implementation for hypertension management in COVID-19 pandemic.**

| row | Unique concept | Initial theme | Final theme |
|-----|----------------|---------------|-------------|
| 1 | A significant reduction in SBP for telemedicine group [10, 32, 34] | Systolic Blood Pressure control [4, 5, 10, 11, 14, 29–34, 36, 37, 40] | BP control [3–6, 10–12, 14, 26, 28–34, 36–38, 40, 41] |
| 2 | During confinement, SBP gradually decreased [5, 29, 30] | | |
| 3 | SBP values were lower in the telemedicine patients [11] | | |
| 4 | Reductions in SBP across all patients were determined [14] | | |
| 5 | SBP achieved statistical and clinical significance [40] | | |
| 6 | The similar mean SBP before and during pandemic [36] | | |
| 7 | SBP did not change during COVID-19 pandemic [37] | | |
| 8 | No significant changes in systolic blood pressure [4, 31] | | |
| 9 | SBP goals was reached [33] | | |
| 10 | SBP were lower in the telemedicine patients [11] | | |
| 11 | During confinement, DBP gradually decreased [5, 30] | Diastolic Blood Pressure control [4, 5, 10, 11, 14, 30, 32–34, 36, 40] | |
| 12 | DBP values were lower in the telemedicine patients [11, 32, 34] | | |
| 13 | Reductions in DBP across all patients were determined [14] | | |
| 14 | No significant difference was observed in the DBP averages between the pre pandemic and the pandemic periods [4] | | |
| 15 | DBP achieved statistical and clinical significance [40] | | |
| 16 | For DBP there was no statistically significant difference [10] | | |
| 17 | The mean DBP increased in type 2 diabetes mellitus [36] | | |
| 18 | SBP goals was reached [33] | | |
| 19 | DBP values were lower in the telemedicine patients [11] | | |
| 20 | Telehealth significantly contributed to the return of control rates to pre-pandemic levels [26] | Total BP control [3, 6, 11, 12, 14, 26, 28–30, 38, 41] | |
| 21 | 24-hour BP control improved [6] | | |
| 22 | Enhanced day-time BP control [6] | | |
| 23 | Nocturnal BP control worsened [6] | | |
| 24 | Telepharmacy visits had a non-significant change in BP control [3] | | |
| 25 | Individuals who had higher baseline BP had a larger decrease [14] | | |
| 26 | Telehealth patients were more likely to have lower BP levels among patients treated with antihypertensive medications [11] | | |
| 27 | There was no significant change detected between the alterations in BP of the intervention group before the pandemic [28] | | |
| 28 | No significant differences in BP control were found between the baseline and after the expansion of telehealth services [38] | | |
| 29 | More decrease was for participants with higher BP [30] | | |
| 30 | No significant differences in BP reduction by the number of visits [14] | | |
| 31 | Most patients treated via telehealth maintained pre-COVID BP [12] | | |
| 32 | Increased telemedicine visit use is associated with poorer performance on the controlling hypertension quality measure [41] | | |
| 33 | Significant improvements in meeting guideline-recommended targets for BP [29] | | |
| 34 | Age had no significant effect on BP changes [14, 30] | Impact of disparities on telehealth outcomes [14, 30, 36] | |
| 35 | Only patients younger than 65 had an increase in DBP [36] | | |
| 36 | Sex had no significant effect on BP changes [14, 30] | | |
| 37 | Male patients, a small not significant decrease in SBP was noted [36] | | |
| 38 | Male patients, a small not significant increase in DBP was noted [36] | | |
| 39 | No significant differences in BP reduction by geographic location [14] | | |

*(Continued)*

**Table 2.** (Continued)

| row | Unique concept | Initial theme | Final theme |
|---|---|---|---|
| 40 | No significant difference between the mean error rate per participant between the face-to-face and remote ABPM cohorts [25] | remote BP measurement [18, 25, 38] | Measurement & recording [17, 18, 25, 34, 38, 41] |
| 41 | Better screening of BP and pre-hypertension [18] | | |
| 42 | The high rate of successful remote BP measurement [25] | | |
| 43 | Less BP assessment among telemedicine encounters [38] | | |
| 44 | Telemedicine visits are less likely to have recorded BP values [41] | Recording of BP [17, 34, 41] | |
| 45 | Documentation of measured BP led to better BP control [34] | | |
| 46 | High rate of using a digital home BP monitor [17] | | |
| 47 | Telemedicine may not negatively impact BP control when BP is recorded [41] | | |
| 48 | There was no difference found for medication adherence between in person pharmacy visits compared to Telepharmacy visits [3] | Medication adherence [6, 11, 12, 17, 29–31, 34] | Medication management [3, 6, 11, 12, 17, 29–31, 34] |
| 49 | Telemedicine medication management had led to better medication adherence improvement than UC [31, 34] | | |
| 50 | increased medication adherence during the COVID-19 pandemic [12] | | |
| 51 | Increased treatment compliance [30] | | |
| 52 | Patients contacted from remote were also less frequently treated with antihypertensive drugs [11] | | |
| 53 | The proportion of patients treated with antihypertensives did not significantly vary during and after the lockdown [6] | | |
| 54 | Adherence to the Mediterranean diet improved [29] | | |
| 55 | The intervention group had a higher level of medication delivery [17] | | |
| 50 | Medication adherence declined comparing to pre COVID-19 year [12] | Drug side effects [3, 12, 31, 34] | |
| 56 | There was no difference found for antihypertensive adverse events between in person pharmacy visits compared to Telepharmacy visits [3] | | |
| 57 | No significant changes in adverse drug events [31] | | |
| 58 | No significant changes in BP symptoms [31] | | |
| 59 | Telehealth group experience less major adverse events [34] | | |
| 60 | Decreased occupational stress level [30] | Improved mental health [29, 30, 40] | mental health [29, 30, 40] |
| 61 | Mental well-being achieved statistical and clinical significance [40] | | |
| 62 | Anxiety and depression levels reduced significantly [29] | | |
| 63 | Access to health services without applying to the hospital [4, 14] | Access to care [4, 14, 31, 34, 39] | Care continuity [4, 14, 31, 34, 39] |
| 64 | Less need to revisits for telehealth group [34] | | |
| 65 | No significant difference in emergency visits between in patient and telemedicine visits [31] | | |
| 66 | The participants stated that their follow-up continued [4] | | |
| 67 | Higher care continuity [39] | | |

(*Continued*)

**Table 2.** (Continued)

| row | Unique concept | Initial theme | Final theme |
|---|---|---|---|
| 68 | The majority of patients started using telehealth [35] | Telemedicine use [1, 12, 14, 17, 24, 26, 31, 35, 38, 39] | Use & acceptance [1, 3, 4, 12, 14, 17, 24, 26, 27, 31, 32, 35, 38–40] |
| 69 | The intervention group had received at least one health coaching session in the last three months [17] | | |
| 70 | Patients with hypertension were more likely to use telemedicine [1] | | |
| 71 | Telehealth use rate increased [12, 24, 26, 31, 38, 39] | | |
| 72 | Telemedicine offers a patient centered tool to better BP management [14] | | |
| 73 | Patients with hypertension had greater odds of having their encounter be through telemedicine [1] | | |
| 74 | Telemedicine in cardiology widely accepted by patients [35] | Telemedicine acceptance [27, 35, 40] | |
| 75 | Virtual visits presented high acceptability [27] | | |
| 76 | The usage of zoom to access the program is acceptable [40] | | |
| 77 | Telemedicine was very effective [14, 35] | Effectiveness [14, 27, 35] | |
| 78 | Virtual visits were easy to be carried out [27] | | |
| 79 | Telemedicine in cardiology was highly feasible [35] | | |
| 80 | The patients were satisfied with telehealth [27, 32] | Patient satisfaction [4, 27, 31, 32] | |
| 81 | The patients preferred telemedicine [31] | | |
| 82 | The patients said that they felt safe in this system [4] | | |
| 83 | Age was not associated with telemedicine use [1, 3] | Impact of disparities on telemedicine use [1, 3, 24] | |
| 84 | Sex was not associated with telemedicine use [1, 3] | | |
| 85 | Less utilization of telemedicine by refugee [1] | | |
| 86 | Non—English patients had lower odds of a telemedicine encounter [1] | | |
| 87 | No racial differences in the use of telemedicine [24] | | |
| 88 | no association between the use of telemedicine and geographic regions [24] | | |

proprietary systems, blood pressure control results have been reported. This rate reaches 54.5% and 44.4% in studies using public and telephone systems, respectively. In a similar vein, all three studies on mental health used proprietary systems.

On the other hand, more studies used telephone calls than proprietary platforms for measuring and recording data (50% vs. 33.3%). Proprietary platforms are more effective for managing blood pressure because they offer functionalities that are specifically designed for disease management. They can also measure blood pressure more accurately. In the context of the special conditions caused by quarantine and social distancing, simpler mechanisms such as public platforms and even telephone calls have also had a positive effect on blood pressure management. Studies have shown that basic telemedicine visits are as effective as face-to-face visits in controlling hypertension [45].

Of the 29 studies analyzed, 6 articles (20.68%) focused solely on hypertension control, while 23 articles (79.31%) examined hypertension in conjunction with other diseases. This suggests that, to a large extent, hypertension is often considered as a comorbid condition. However, elevated hypertension can be a fundamental cause of other serious conditions, such as heart disease and stroke.

It is important to note that the key factor in successfully replacing face-to-face visits with telemedicine is patient training [46]. In other words, the higher the quality of patient training, the greater the likelihood of telemedicine success. Almost half of the studies (N = 13, 44.8%) provided necessary training, such as instructions on how to use wearable devices or home hypertension measuring devices. The participants received instructions on how to communicate clinical information to physicians or support staff, as well as lifestyle modification methods to help manage their hypertension. Most of studies (N = 11, 37.9%) used virtual methods

such as videos, workshops, books, articles, websites, and emails for training. These studies obtained significantly more useful results than those that did not educate patients or their companions.

In light of the rising prevalence of hypertension and the constraints of conventional care models, the implementation of novel interventions in patients' homes may facilitate enhanced accessibility, quality, and outcomes [47].

This study identified six main themes regarding the outcomes of using telemedicine for hypertension management.

## BP control

Several studies have shown that telemedicine is a pragmatical solution for managing BP. this systematic review indicates that telemedicine is not only effective, but also incredibly successful in reducing or maintaining BP (both systolic and diastolic) in most studies. Only three studies reported worsening BP management, which was largely influenced by study setting and design. One study found that diabetic patients experienced an increase in diastolic BP [36], while Omboni reported a worsening of nocturnal BP. It is worth noting that this study was the only one to examine blood pressure separately during the day and at night [6]. It was also reported that there was a decrease in the qualitative measures of BP [41]. However, if the data for BP measurement is recorded correctly, providing care through telemedicine should not have a negative impact on hypertension management [41]. During pandemics, such as the COVID-19 outbreak, quarantine measures can disrupt screening services and care, hindering the treatment and control of people with hypertension and making them more susceptible to cardiovascular events [7].

## BP measurement & recording

The effectiveness of a remote BP monitoring depends on accurate, appropriate, and cost-effective BP measurement. Better hypertension measurement and documentation lead to better BP control. The accuracy of blood pressure (BP) measurements is significantly affected by the circumstances under which they are taken. For instance, the stress of being in a hospital during an epidemic can impede the ability to obtain precise BP readings. Telemedicine modalities can address the global demand for a patient friendly approach for BP measurement. As a result, continuous BP measurement and remote monitoring have emerged as a growing field in the health industry [48]. Data recording in telemedicine requires careful attention. Better results are achieved when data is entered automatically. However, if patients have to record the data themselves, it is necessary to perform quality check.

## Medication management

Adherence to medication is crucial in controlling complications of hypertension. Over the past two decades, many countries have seen significant increases in hypertension control rates due to increased access to antihypertensive drug therapy [49]. The most important aspect is adherence to the prescribed medication regimen and its correct usage [15]. Most studies have shown an improvement in adherence to treatment or diet. However, Steiner reported a decrease in adherence in 2020 compared to previous year [12]. During the COVID-19 pandemic, medication adherence has increased due to the availability of mail-order delivery and ≥90-day medication supplies. Furthermore, adherence to treatment has resulted in fewer drug side effects.

Telemedicine has improved access to antihypertensive drugs for patients in remote areas. Some governments, including China and the United States, provide affordable health

insurance and antihypertensive drugs [50]. Mobile health technologies provide patients with a self-management framework to enhance medication adherence through physiological data monitoring and timely alerts and reminders [51]. Automated summary reports of patient adherence and hypertension can be easily uploaded to provider-connected networks to help reduce clinical inertia [51]. Telemedicine offers several tools to prevent accidental forgetfulness and non-adherence to medication, including drug interaction warnings, medication reminders, and BP check reminders. [52].

## Mental health

Telemedicine has been shown to alleviate stress and anxiety among patients and staff by eliminating the fear of infection during face-to-face appointments at the office or hospital. Other studies have also reported a reduction in stress and anxiety as a result of using telemedicine during the COVID-19 pandemic [53, 54]. While depression and anxiety are risk factors for cardiovascular disease, their relationship with HBP is less clear. However, that relationships may be confounded by factors related to age and lifestyle [55]. Generally, exposure to remote treatment and control can reduce stress and anxiety and increase willingness to cooperate.

## Care continuity

The use of telemedicine has increased access to screening, diagnosis, and primary treatment services without the need for in-person visits. It has also made it possible for patients to receive remote follow-up care and renew their prescriptions. Telemedicine has made a significant contribution to the continuity of care for hypertension patients by covering the full range of care. The importance of continuity in hypertension management has already been well demonstrated. Advancements in technology facilitate the continuity of care and enable flexible performance in process therapeutics. The impact of continuity of care on the wider adoption of telehealth should be considered to inspire new patient-centered innovations [39].

## Use & acceptance

Telemedicine has gained considerable traction in various medical fields in recent years [56]. This is largely attributable to the pervasiveness of the internet and mobile phones. One of the key considerations in the use of telemedicine is the availability of an adequate communication infrastructure. According to this systematic review study, telemedicine is frequently used for hypertension management in countries with high rates of internet and mobile phone penetration. The second significant factor in the use of telemedicine is the clinical field. The findings indicated that the management of hypertension during the COVID-19 pandemic has been more prevalent than that of other chronic diseases. One reason for this is the relative ease of measuring high blood pressure data compared to other conditions such as diabetes and or kidney diseases. While Telemedicine is used to treat a variety of heart diseases, it is particularly beneficial for patients with hypertension [1] due to its ease of management compared to other heart conditions.

While other studies have extensively investigated the impact of differences such as race, age, and literacy on the use of information technology [57], the reviewed studies have addressed these issues less and they were not among their primary goals. Only one study demonstrated a lower use of telemedicine by refugees and non-English speaking patients.

Telemedicine provides a patient-centered approach for hypertension patients to access care. Currently, there is a growing call from patient advocates, policymakers, and healthcare leaders for greater patient involvement in shaping the healthcare landscape. Telemedicine is widely accepted by physicians and patients [58]. In the majority of studies, patients and

medical staff reported satisfaction with using telehealth for treatment, follow-up, or disease prevention.

## 5. Limitation

The study has three limitations. Firstly, there was a lack of detailed examination of changes in systolic and diastolic blood pressure due to the heterogeneity and differing methodologies of the studies, also the review was not registered in Prospero. Secondly, the study did not aim to provide a detailed examination of the factors that affect the use of telemedicine tools, such as their level of complexity or connection with health information systems. Future studies could provide a new perspective on the use of these tools in hypertension management by conducting a detailed investigation of these factors. Additionally, the number of RCTs was limited, which may have impacted the strength of the evidence in this area. Despite their stronger methodology, RCTs were difficult to conduct due to the isolation and social distancing measures imposed by COVID-19.

## 6. Conclusion

Telemedicine allows patients with hypertension to have medical consultations more conveniently and comfortably and considered as valid as in-person visits. This study reports the outcomes of using telemedicine for BP management including BP control, BP measurement and recording, medication management, mental health, care continuity, and use and acceptance. Telemedicine offers the possibility of creating a connected network to support patients with high BP anywhere and anytime. High BP requires regular treatment and monitoring of BP, as well as adherence to physician's recommendations. Limitations and issues may arise due to patients and healthcare staff being unfamiliar with telemedicine. These problems can be resolved through ongoing use and continuous feedback.

## Supporting information

**S1 Checklist. PRISMA 2020 checklist.**
(DOCX)

**S1 Appendix. Search strategy.**
(DOCX)

**S2 Appendix. Quality appraisal results.**
(DOCX)

## Author Contributions

**Conceptualization:** Abdullah Gharibzade, Niloofar Choobin, Haniyeh Ansarifard.

**Data curation:** Mohammad Hosein Hayavi-haghighi, Abdullah Gharibzade, Niloofar Choobin, Haniyeh Ansarifard.

**Formal analysis:** Mohammad Hosein Hayavi-haghighi, Niloofar Choobin, Haniyeh Ansarifard.

**Investigation:** Mohammad Hosein Hayavi-haghighi, Haniyeh Ansarifard.

**Methodology:** Mohammad Hosein Hayavi-haghighi, Abdullah Gharibzade, Haniyeh Ansarifard.

**Project administration:** Mohammad Hosein Hayavi-haghighi, Haniyeh Ansarifard.

**Resources:** Mohammad Hosein Hayavi-haghighi, Haniyeh Ansarifard.

**Software:** Mohammad Hosein Hayavi-haghighi, Haniyeh Ansarifard.

**Supervision:** Mohammad Hosein Hayavi-haghighi, Abdullah Gharibzade, Haniyeh Ansarifard.

**Validation:** Mohammad Hosein Hayavi-haghighi, Niloofar Choobin, Haniyeh Ansarifard.

**Visualization:** Mohammad Hosein Hayavi-haghighi, Haniyeh Ansarifard.

**Writing – original draft:** Mohammad Hosein Hayavi-haghighi, Abdullah Gharibzade, Niloofar Choobin, Haniyeh Ansarifard.

**Writing – review & editing:** Mohammad Hosein Hayavi-haghighi, Niloofar Choobin.

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
