## [Editor Report · Decision Letter 0]

8 Apr 2024

PONE-D-24-11118Applications and outcomes of implementing telemedicine for hypertension management in COVID 19 pandemic: a systematic reviewPLOS ONE

Dear Dr. ansarifard,

Thank you for submitting your manuscript to PLOS ONE. While I think your article is potentially interesting to our readers, I don't think that I can send this article for external peer-review in its current state. I am using this revision decision to give you an opportunity to improve the quality of your manuscript before further consideration with external peer-reviewers. Important items of note that need to be edited include:- Overall use of English. Please ensure that you are using academic language; do not use conversational phrasing such as "dealt with" or "which stands for". I think this manuscript would greatly benefit from edits from an English native author, or from a language service. Please DO NOT edit English using AI-powered services, such as ChatGPT.- Please make sure that your spelling and abbreviations are consistent (such as spelling COVID-19 correctly, and not "Covid 19". Make sure all of your sentences and paragraphs are complete (such as incomplete sentence on line 133).- The search strategies on appendix 1 is woefully insufficient. Apart from significant spelling errors (such as "Cochran", and "Web of Since"), make sure that a separate table is provided for each database, with **replicable search statements** (i.e., if the statement is entered into the platform, it would replicate the author's search exactly). At the moment, I have no idea what exact strategy the authors used.- Authors also need to review their systematic review methodology knowledge and ensure that PRISMA 2020 is followed correctly. For instance, the authors used the CASP framework for item 14 on PRISMA 2020. This is incorrect; item 14 is referring to publication bias, and should be evaluated appropriately. Please review the checklist and revise your methodology. Same for item 15. Quality of evidence is different from per-study risk of bias assessments.- The authors did not append a PRISMA flowchart, as required by the PRISMA guideline. Overall, this review shows a general lack of care for manuscript quality and systematic review methodology despite the interesting topic. The authors need to review their methods extensively, otherwise I would have to unfortunately reject this manuscript without review.

We look forward to receiving your revised manuscript.

Kind regards,

Jiawen Deng

Academic Editor

PLOS ONE

2. Please note that PLOS ONE has specific guidelines on code sharing for submissions in which author-generated code underpins the findings in the manuscript. In these cases, all author-generated code must be made available without restrictions upon publication of the work. Please review our guidelines at https://journals.plos.org/plosone/s/materials-and-software-sharing#loc-sharing-code and ensure that your code is shared in a way that follows best practice and facilitates reproducibility and reuse."

---

## [Author Response · Author response to Decision Letter 0]

24 Apr 2024

In accordance with the instructions set forth in the Response to Reviewers file, the comments received have been carefully considered and numerous revisions have been implemented to the manuscript in light of them.

---

## [Decision Letter · Decision Letter 1]

23 May 2024

PONE-D-24-11118R1Applications and outcomes of implementing telemedicine for hypertension management in COVID 19 pandemic: a systematic reviewPLOS ONE

Dear Dr. ansarifard,

Thank you for submitting your manuscript to PLOS ONE. After careful consideration, we feel that it has merit but does not fully meet PLOS ONE’s publication criteria as it currently stands. Therefore, we invite you to submit a revised version of the manuscript that addresses the points raised during the review process.

We look forward to receiving your revised manuscript.

Kind regards,

Josep Vidal-Alaball, MD, PdD, MPH

Academic Editor

PLOS ONE

Journal Requirements:

Reviewers' comments:

Reviewer's Responses to Questions

**Comments to the Author**

1. If the authors have adequately addressed your comments raised in a previous round of review and you feel that this manuscript is now acceptable for publication, you may indicate that here to bypass the “Comments to the Author” section, enter your conflict of interest statement in the “Confidential to Editor” section, and submit your "Accept" recommendation.

Reviewer #1: All comments have been addressed

Reviewer #2: (No Response)

2. Is the manuscript technically sound, and do the data support the conclusions?

Reviewer #1: Yes

Reviewer #2: Yes

3. Has the statistical analysis been performed appropriately and rigorously? 

Reviewer #1: Yes

Reviewer #2: I Don't Know

4. Have the authors made all data underlying the findings in their manuscript fully available?

Reviewer #1: Yes

Reviewer #2: Yes

5. Is the manuscript presented in an intelligible fashion and written in standard English?

Reviewer #1: Yes

Reviewer #2: Yes

6. Review Comments to the Author

Reviewer #1: Main comments:

The review article explores telemedicine applications and outcomes in monitoring patients with hypertension amidst the Covid-19 pandemic, highlighting existing opportunities and challenges in telemedicine deployment for managing these patients during the Covid-19 crisis. This significant work has the potential to catalyze telemedicine implementation for the monitoring and management of hypertension patients.

The manuscript acknowledges the diversity in study methods, participant demographics, and variations in telemedicine deployment outcomes in the context of Covid-19, all of which contribute to the heterogeneous quality of evidence. A more detailed examination of these factors could shed further light on their impact on study outcomes and the extent to which these results are generalizable to other settings and chronic diseases. Analyzing how these variables affect outcomes could enhance our understanding of the collected evidence. While the article presents various telemedicine implementation strategies, there is room for further exploration of how these differences may impact monitoring outcomes for these patients.

Positive points:

• The authors correctly structured the PRISMA 2020 method, as referenced, to conduct the review by selecting studies in accordance with the guidelines.

• The results highlighted several challenges related to telemedicine deployment in different countries, such as the development of suitable software and internet access.

• The introduction is well-structured, providing a brief definition of telemedicine, the Covid-19 context, and the necessity of remote monitoring for hypertension patients, particularly with blood pressure (BP) monitoring.

• In the methodology section, the inclusion and exclusion criteria, as well as the search strategy, were well established. Four databases were identified, yielding a sufficient number of 29 articles for the review, with robust criteria set by the Critical Appraisal Skills Programme (CASP) for the quality of peer-reviewed studies.

• The review tables are comprehensive, detailing the characteristics, outcomes, and results found in each observed study and corresponding to the cited references.

• In the discussion, the author provides relevant comparisons with the main study outcomes, clearly addressing how telemedicine implementation can contribute to hypertension patient monitoring, and mentioning the limitations found in the review.

Negative points:

• The authors did not explicitly address potential biases in the selected studies.

• The heterogeneity and differing methodologies among the studies were not adequately addressed.

• Factors affecting the use of telemedicine tools were not discussed in detail.

• As a study limitation, the review was not registered in Prospero.

• In line 93, correct spacing between words and brackets: "ability to effectively manage their hypertension over the past decade (14)."

• In line 133, correct spacing between words: "OR SARS-COV-2 OR 2019 Novel Coronavirus. We then determined."

• In line 292, correct spacing between words: "in 2020 compared to the pre-COVID 19 year (2019)."

I recommend publishing the article on plosone with the necessary changes

Reviewer #2: This is a well written systematic review with appropriate methodology and transparent process of selection of articles. The discussion is nicely divided into subheadings and required prospects of telemedicine are covered.

There is just one suggestion- Kindly add a paragraph on the challenges being faced during Telemedicine consultations other than inappropriate patient training.

7. PLOS authors have the option to publish the peer review history of their article (what does this mean?). If published, this will include your full peer review and any attached files.

Reviewer #1: **Yes: **João Paulo Batista de Souza - Federal University of Latin American Integration

Reviewer #2: No

---

## [Author Response · Author response to Decision Letter 1]

11 Jun 2024

Reviewer #1: 

Main comments:

We would like to express our gratitude for your kind remarks regarding our manuscript. However, we would like to draw your attention to the following:

o Heterogeneity of evidence:

The heterogeneity of evidence is most frequently addressed through meta-analysis studies. However, given the constraints of time and resources, we were compelled to limit our efforts to a systematic review. This decision was influenced by several considerations, including the difficulty in obtaining a statistical expert and the paucity of time. This issue has been identified as the first limitation of the research.

o While the article presents various telemedicine implementation strategies, there is room for further exploration of how these differences may impact monitoring outcomes for these patients.

We would like to thank you for your constructive comment. Please refer to lines 235-258 on pages 13-14 for further details.

Negative points:

• The authors did not explicitly address potential biases in the selected studies.

The risk of bias assessment is a more common practice in meta-analysis studies, whereas in systematic reviews, quality appraisal is typically employed. In this study, we employed the CASP methodology to quality appraisal of the included studies. The results of this appraisal are presented in Appendix 2.

• The heterogeneity and differing methodologies among the studies were not adequately addressed.

Heterogeneity is an essential element in meta-analysis research. However, for reasons previously stated, this aspect was not included in our objectives and therefore not included in our study.

• Factors affecting the use of telemedicine tools were not discussed in detail.

The necessary edition was implemented in accordance with your comment in lines 338-343, page 16.

• As a study limitation, the review was not registered in Prospero.

This limitation was subsequently incorporated into the limitations of the study, page 16 in line 359

• In line 93, correct spacing between words and brackets: "ability to effectively manage their hypertension over the past decade (14)."

• In line 133, correct spacing between words: "OR SARS-COV-2 OR 2019 Novel Coronavirus. We then determined."

• In line 292, correct spacing between words: "in 2020 compared to the pre-COVID 19 year (2019).

We would like to extend our apologies for the errors that have been identified in the manuscript. All errors in the text have now been rectified.

Reviewer #2:

There is just one suggestion- Kindly add a paragraph on the challenges being faced during Telemedicine consultations other than inappropriate patient training

We would like to express our gratitude for your kind comments and your profound generosity regarding our submitted manuscript. Furthermore, we have considered your valuable suggestion and have incorporated additional items into the manuscript on page 4, lines 108-114.

---

## [Decision Letter · Decision Letter 2]

15 Jun 2024

Applications and outcomes of implementing telemedicine for hypertension management in COVID 19 pandemic: a systematic review

PONE-D-24-11118R2

Dear Dr. ansarifard,

We’re pleased to inform you that your manuscript has been judged scientifically suitable for publication and will be formally accepted for publication once it meets all outstanding technical requirements.

Kind regards,

Josep Vidal-Alaball, MD, PdD, MPH

Academic Editor

PLOS ONE

Additional Editor Comments (optional):

The reviewers' concerns have been correctly addressed.

Reviewers' comments:

Reviewer's Responses to Questions

**Comments to the Author**

1. If the authors have adequately addressed your comments raised in a previous round of review and you feel that this manuscript is now acceptable for publication, you may indicate that here to bypass the “Comments to the Author” section, enter your conflict of interest statement in the “Confidential to Editor” section, and submit your "Accept" recommendation.

Reviewer #1: All comments have been addressed

2. Is the manuscript technically sound, and do the data support the conclusions?

Reviewer #1: Yes

3. Has the statistical analysis been performed appropriately and rigorously? 

Reviewer #1: N/A

4. Have the authors made all data underlying the findings in their manuscript fully available?

Reviewer #1: Yes

5. Is the manuscript presented in an intelligible fashion and written in standard English?

Reviewer #1: Yes

6. Review Comments to the Author

Reviewer #1: I appreciate the time and effort you dedicated to responding to the comments and suggestions provided.

I am pleased to inform you that your revised manuscript has satisfactorily addressed the concerns raised during the review process. The additional discussions on the heterogeneity of evidence, potential biases, differences in methodologies, and factors affecting the use of telemedicine have significantly strengthened the clarity and comprehensiveness of your study. Additionally, the corrections to the formatting errors have been duly noted and appreciated.

Given these improvements, I am pleased to accept your manuscript for publication in PLOS ONE. Congratulations on your excellent work. I look forward to seeing your valuable contribution published.

7. PLOS authors have the option to publish the peer review history of their article (what does this mean?). If published, this will include your full peer review and any attached files.

Reviewer #1: **Yes: **João Paulo Batista de Souza

Federal University of Latin American Integration (UNILA)

---

## [Editor Report · Acceptance letter]

28 Jun 2024

PONE-D-24-11118R2 

PLOS ONE

Dear Dr. ansarifard, 

I'm pleased to inform you that your manuscript has been deemed suitable for publication in PLOS ONE. Congratulations! Your manuscript is now being handed over to our production team.

Kind regards, 

on behalf of

Dr. Josep Vidal-Alaball 

Academic Editor

PLOS ONE